# The Association between Parent and Child-Report Measures of Alexithymia in Children with and without Developmental Language Disorder

**DOI:** 10.3390/ijerph18168309

**Published:** 2021-08-05

**Authors:** Hannah Hobson, Neeltje P. van den Bedem

**Affiliations:** 1Department of Psychology, University of York, York YO10 5DD, UK; 2Developmental Psychology, Leiden University, 2311 EZ Leiden, The Netherlands; n.p.van.den.bedem@fsw.leidenuniv.nl

**Keywords:** developmental language disorder, language impairment, emotion, alexithymia

## Abstract

Accurate measures of alexithymia, an inability to recognise and describe one’s own emotions, that are suitable for children are crucial for research into alexithymia’s development. However, previous research suggests that parent versus child reports of alexithymia do not correlate. Potentially, children may report on the awareness of their emotions, whereas parent-report measures may reflect children’s verbal expression of emotion, which may be confounded by children’s communicative abilities, especially in conditions such as Developmental Language Disorder (DLD). Given theoretical arguments that alexithymia may develop due to language impairments, further research into alexithymia in DLD is also needed. This project examined parent and child report measures of alexithymia in children with DLD (*n* = 106) and without DLD (*n* = 183), and their association to children’s communication skills. Parent and child reports were not significantly correlated in either group, and children with DLD had higher alexithymia scores on the parent-report measure only. Thus, parent and child measures of alexithymia likely reflect different constructs. Pragmatic language problems related to more parent-reported alexithymia, over and above group membership. Structural language abilities were unrelated to alexithymia. We suggest decreased social learning opportunities, rather than a language measure artefact, underlie increased alexithymic difficulties in DLD.

## 1. Introduction

Alexithymia is a personality construct that encapsulates difficulties recognising and expressing one’s own affect. Children with alexithymic traits may experience negative arousal but be unable to correctly link their feelings to the cause of their emotion, or they may have difficulties expressing what they feel in a nuanced manner. This has a negative effect on their ability to regulate and express their emotions in an adaptive way [1]. Alexithymia has been shown to be elevated in a range of clinical populations [2,3,4], and to be associated with poorer socioemotional functioning [5,6,7] and poorer mental wellbeing [8,9]. Given these associations, a clear priority for researchers of alexithymia is understanding what leads to its development.

One practical difficulty in investigating the development of alexithymia is how this construct is best measured in children. In adult research, alexithymia is usually measured via a self-report questionnaire, and both measures using self-report and observer-report (typically administered to parents) are available for use with developing samples. However, there has been little research on the agreement between self-report and parent-report measures of alexithymia, and the small number of studies that has been published suggest non-significant or weak correlations between parent and child reports of alexithymia. Griffin, Lombardo, and Auyeung [10] collected self-report and parent-report data from autistic children and typically developing controls. For each group separately, they examined the correlation between their parent-report and child-report measures of alexithymia: these measures did not significantly correlate in either group (although the small sample size and corrected significance threshold would arguably have meant only a large correlation would have been detected). Similarly, Hobson et al. [11] reported the association between parent-reported alexithymia and child-reported alexithymia for autistic adolescents or adolescents at genetic risk of autism. The measures were significantly correlated but weakly (*r* = 0.19), given these measures purportedly reflect the same construct.

Weak or non-significant associations between parent and child report measures of alexithymia may suggest several things. Firstly, young children may lack the meta-cognitive awareness needed to reliably report their own emotional abilities. Participants in Griffin et al.’s [10] study were aged between 8 and 13 years; the youngest children in this sample may have found reflecting on their own emotional abilities difficult. However, previous research on parent–child agreement for other emotion-related constructs have not shown age effects: children’s age (for a sample aged between 9 and 13 years) did not affect the agreement between child and parent reports of children’s anxiety symptoms [12]. Furthermore, child rather than parent reports have sometimes been more predictive of meeting diagnostic outcomes [12,13]. Thus, we should be careful in assuming the veracity of parent report over children’s reports of their own emotional experiences. This brings us to the second explanation for weak correlations between reports: perhaps parents are not accurate reporters of their children’s true alexithymic traits. Arguably, the emotional difficulties considered by these measures are too private for anyone other than the individual themselves to reliably report.

It remains unclear which account, child or parent-report, provides a more accurate reflection of children’s emotional abilities. Some findings may reflect the particular samples used across these studies; possibly for some populations, such as children and adolescents with autism (as in the samples of Griffin et al. [10] and Hobson et al. [11]), insight into own emotional abilities in disrupted. Another possibility is that agreement depends on the factor at hand: Cantwell et al. [14] reported that agreement was good for many externalising symptoms, such as those associated with conduct disorder, but agreement was weaker for internalising problems.

Rather than attempting to unpick which report is “right”, a third possibility is that parents and children are using different sources of information to judge children’s emotional insight, and that either report may be more reflective of certain aspects of the alexithymic construct. Specifically, parent-report measures of alexithymia may be particularly reliant on children’s verbal expressions of their own emotions. Indeed, during the development of the parent-report measure of alexithymia, the Children’s Alexithymia Measure (CAM), Way et al. [15] sought items that reflected externally observable behaviours, the majority of which appear to reflect problems with emotional expression, for example: [my child] “Uses few words (may just say “good”/”bad”) to describe most of his/her feelings” or [my child] “Says “I don’t know” when asked why he/she is upset”. In fact, only one item in the CAM does not make explicit reference to the child’s verbal expressive abilities. Difficulty expressing feelings is a key element of the alexithymia concept, but if parent-report measures reflect predominantly this factor then this will miss other key aspects of alexithymia, such a difficulty recognising one’s own emotions, understanding the cause of the emotion, and having an externally oriented thinking style. This would also weaken the association between parent and child reports.

If parent-report measures rely heavily on verbal expression of emotions, this also opens up the possibility that such measures will be particularly affected where children have a broader communication problem. There is some previous evidence that verbal abilities may affect alexithymia ratings. In the validation paper for the CAM; communication impairments were found to affect CAM scores, with children with a history of communication problems showing significantly higher alexithymia [16]. An association between verbal abilities and alexithymia could reflect a causal role for language skills in emotional abilities. Language problems have been argued to directly contribute to alexithymia [17,18], drawing on constructionist accounts of emotional development [19] that would predict that language problems derail the learning of emotional concepts, leading to problems with emotion recognition and regulation. Alternatively, language problems may indirectly lead to alexithymia, as communication difficulty may lead to increased social exclusion and diminished friendship quality, reducing opportunities to develop socioemotional skills, including emotion recognition in oneself and in others [20]. This model has been described as a “transactional cycle of interaction” (see [21]). As yet, it is not clear what aspects of language and communication problems are associated with alexithymia, though constructionist theories of emotional development would seem to highlight structural language skills (i.e., lexical and syntactic skills), especially vocabulary, as important [22]. However, recent investigations with populations with acquired communication problems (following stroke) have suggested that pragmatic rather than structural language skills may be associated with alexithymia [23].

Associations between alexithymia scores and children’s communication problems could also reflect simple measurement confound. Alexithymia measures aim to capture difficulty identifying and expressing emotions, but children with communication problems will have general expression difficulties, affecting multiple topics, not just emotion. For example, finding it hard to put your feelings into words may reflect general word finding difficulties, having very few words to describe your emotional experiences may reflect broader non-specific vocabulary problems, and talking about unrelated topics when asked about emotions may be due to underlying comprehension difficulties. Parents of children with language problems and language-impaired children themselves completing questionnaires that ask about their children’s expression of their emotions will therefore suggest difficulties, but in reality these problems are not specific to emotional abilities.

Overall, there has been limited study of the agreement between child and parent reports of alexithymia, a knowledge gap that is important to address, in order for developmental researchers to have confidence in these measures when seeking to study the ontogeny of alexithymia and emotion processes. The potential role of language problems in measuring alexithymia in developing samples has also had little consideration, despite current theories suggesting a “language route” to alexithymia [17,18].

The current project investigated the agreement between child self-report and parent-report measures of alexithymia. We also examined the association between alexithymia and communication difficulties. We did this using both a group-based and continuous measure approach: we compared children with Developmental Language Disorder (DLD) to their typically developing peers, and examined the correlations between alexithymia measures and continuous measures of communication ability, which included measures of both structural language and pragmatic language. Specifically, we tested: (a) whether parent and child report measures would correlate, for children with and without DLD; (b) whether children with DLD score higher on measures of alexithymia compared to children without DLD; (c) whether pragmatic versus structural language problems would be associated with alexithymia in children with and without DLD; (d) whether measures of alexithymia that show relationships to language abilities show a similar factor structure for children with and without DLD.

## 2. Materials and Methods

### 2.1. Participants

The sample consisted of 183 typically developing children and 106 children with a diagnosis of DLD. Children with DLD were recruited via schools for children with DLD, or through specialised organisations aimed at supporting children with DLD. Children with DLD were recruited between September and December 2014, in which the DSM-IV was used in the Netherlands. To be included in the DLD sample, children had to have received a diagnosis of DLD, according to the DSM-IV criteria. None of the DLD sample had been diagnosed with autism spectrum disorder, or been shown to have a hearing impairment. Typically developing children were recruited via mainstream schools. This sample formed part of an ongoing programme of research and has been reported on in previous publications [18,20,24,25,26].

The majority of both groups of children had one or both parents who originated from the Netherlands (With DLD: 81% from the Netherlands, 6% from other countries, 13% missing data; without DLD: 74% from the Netherlands, 4% from other countries; 22% missing data). Table 1 summarises the sample’s characteristics. The two groups were equivalent in terms of their age, but did significantly differ on socioeconomic (SES) and performance IQ.

### 2.2. Measures

The parent and child questionnaire measures are described below. Performance IQ (PIQ) also was measured via two subtasks of the Wechsler Intelligence Scale for Children, Third Edition (Block Design and Picture Arrangement; [27]), or when available, PIQ scores were obtained from school or medical files for children with DLD. Socioeconomic status (SES) was estimated via neighbourhood SES, based on families’ postal codes which reflects the mean level of education, income, and occupation of the adults in a neighbourhood compared to all other neighbourhoods in the Netherlands (Mean (*SD*): 0.28 (1.09), and Range = −6.8 to 3.1).

#### 2.2.1. Children’s Alexithymia Measure (CAM)

The CAM [15] is a 14-item parent-report measure of alexithymia of their child. In order to be comparable to the child-report measure, mean scores on the CAM were calculated rather than total scores. Parents responded to the items using a 4-point Likert-scale. Cronbach’s alpha showed responses for this measure were internally reliable for both groups: TD *α* = 0.91, DLD *α* = 0.91.

#### 2.2.2. Emotional Awareness Questionnaire (EAQ)

The EAQ is a child-report measure of their own emotional awareness. The subscale ‘differentiating emotions’ measures whether children are able to recognise and understand the causes of their own basic emotions (DIF), which contains 7 items. Children responded on a 3-point Likert scale. For consistency and ease of interpretation, we scored the EAQ responses to be consistent with the CAM in that higher scores reflect greater alexithymia difficulties. Cronbach’s alpha showed responses for this measure were internally reliable: *α* = 0.74. Examining the groups’ reliability for this self-report measure separately, responses from the children in the TD and DLD groups did not differ in their reliability: TD *α* = 0.72, DLD *α* = 0.78

#### 2.2.3. Children’s Communication Checklist (CCC) (Second Edition)

The CCC is a parent-report measure of children’s communication problems [28,29]. The CCC provides an overall score, in addition to a pragmatic language score, and subscales for structural language areas and speech. Parents responded to 56 questions about speech production, syntax, semantic, coherence, pragmatic abilities (the two scales developed for screening for ASD were not used). Higher scores indicate greater communicative impairment. Parents responded on a 4-point Likert scale to indicate how often communication problems happened. Cronbach’s alpha showed responses for this measure were internally reliable for the general and pragmatic problems (*α* > 0.82), as well as for the separate structural language scales in children with DLD (*α* range between 0.60 and 0.79). However, for the TD group, the structural language subscales were not reliable and were not used in the analyses [29].

### 2.3. Procedure

This study was granted ethical approval by the local ethics committee of Psychology at Leiden University (project 1308277752). Informed consent was given by the parents of the children and by children above 12 years of age. Children completed self-report measures with a trained researcher in a quiet room at school or at home. All questions were read aloud for children with DLD. Children answered the questions privately on a laptop or iPad. Parents completed questionnaires online or via post.

### 2.4. Analytical Approach

We address first the issue of whether parent and child reports of alexithymia agree. We examined Pearson’s correlations between the different reporter measures of alexithymia. We examined these correlations for the whole sample, and for the DLD and TD groups separately.

We then proceed to examine the role of language problems, using both a group-based approach (i.e., comparing the DLD group to the TD group) and continuous factor-based approach (i.e., examining correlations with the CCC scales). For the comparisons between the DLD and TD groups on child self-reported (SR-alexithymia) and parent-reported alexithymia (PR-alexithymia) measures, in order to control for the group differences in SES and PIQ, we conducted a MANCOVA, with SES and PIQ entered as covariates. Following significant group effects, to quantify the amount of variance in alexithymia explained by group membership, over and above SES and PIQ, we conducted hierarchical regression analyses. For our continuous factor approach, we examined the correlations between alexithymia measures and parent-reported communication problems, considering overall communication ability and communication subscale scores, to examine the relative contribution of pragmatic versus structural language problems. Further regressions were conducted to examine whether associations between specific language abilities and alexithymia remained after controlling for background/demographic variables.

Finally, given significant group differences and correlations with language abilities for the parent-reported alexithymia measure, we examined the factor structure of this alexithymia measure, seeking to understand whether the factor structure of this construct indicated multiple factors, some of which might be more reflective of a general language ability, rather than the specific emotional deficit alexithymia is supposed to convey. As we were interested in whether the alexithymia measures would show a factor structure that suggested components reflective of language ability, but we did not have set ideas about how many factors might emerge, we opted to use an Exploratory Factor Analysis approach.

Readers may note that the data reported here formed part of a wider project on DLD [18,20], and thus power calculations for the specific analytical tests detailed here were not conducted prior to data collection: however, we did consider whether our dataset would be sufficient for our main objectives. We posit that to argue that parent and child reports of alexithymia were in agreement, we would expect at least a moderate sized correlation; this would require 88 parent-child datapoints, based on power calculations in G*Power [30], based on 90% power. Our sample size was thus sufficiently large to allow us to explore this separately in the two groups (TD and DLD). Similarly, a MANCOVA comparing the two groups on alexithymic traits, assuming a medium effect and 90% power, would require a total sample size of 171 participants. Thus, we were suitably powered to detect meaningful relationships between parent and child alexithymic reports, and differences between DLD and TD groups.

All tests of significance were two-tailed unless otherwise stated. Analytical tests were conducted using SPSS (Version 26, IBM Corp., Armonk, NY, USA), with additional tests run in R (reference R project), using the package psych, for determining the number of factors in our factor analysis.

## 3. Results

### 3.1. Correlations between Alexithymia Measures

Correlations were considered for the two groups separately as well as across the whole sample, via Pearson correlations. When both groups were analysed together, the correlation between PR-alexithymia and SR-alexithymia scores was not significant: *r* = 0.12, *p* = 0.076 (*n* = 240). For the TD group and DLD group separately, PR-alexithymia scores were also not significantly correlated with SR-alexithymia scores (*r* = 0.099, *p* = 0.227 (*n* = 151) and *r* = 0.10, *p* = 0.35 (*n* = 89), respectively). Partialling out age, SES or PIQ did not change these results. Additionally, no correlations were present when boys and girls were examined separately.

### 3.2. Alexithymia Scores in DLD vs. TD

A MANCOVA was conducted to compare the DLD and TD groups on PR-alexithymia and SR-alexithymia, with covariates PIQ and SES. The Box M test was non-significant indicating the covariance matrices of the dependent variables were equal across groups; homogeneity of regression slopes was examined and again met the assumptions for a (M)ANOVA. The analysis indicated a significant multivariate effect (*F* (2, 132) = 15.96, *p* < 0.001, Wilk’s *Λ* = 0.81). Examining the two alexithymia variables separately, there was a significant group effect on PR-alexithymia (*F* (1, 133) = 31.89, *p* < 0.001), but not SR-alexithymia. (*F* (1, 133) = 0.72, *p* = 0.40). To quantify the amount of variance in PR-alexithymia explained by diagnostic group, after controlling for SES and PIQ, a hierarchical regression was conducted, with SES and PIQ entered in step 1 and diagnostic group in step 2. The addition of step 2 added significant explained variance to the model (*F* change (1, 232) = 47.46, *p* < 0.001). Diagnostic group explained a further 15% of variance in PR-alexithymia scores, over and above SES and PIQ. In Step 1, Both PIQ and SES significantly predicted PR-alexithymia scores (PIQ: *t* = −3.37, *p* = 0.001, Beta = −0.21; SES: *t* = −4.07, *p* < 0.001, Beta = −0.25). In Step 2, SES remained significant: *t* = −2.37, *p* = 0.018, Beta = −0.14. Diagnostic group in step 2 was a significant predictor: *t* = 6.89, *p* ≤ 0.001, Beta = 0.45. VIF (Variance Inflation Factors) were all below 1.4, thus there was no indication of multicollinearity issues.

To examine whether any CAM items in particular drove this group difference in PR-alexithymia scores, we also compared the TD and DLD groups on the scores on individual PR-alexithymia items. The DLD group scored significantly higher on all items (see Table 2).

### 3.3. Associations between Alexithymia and Communication Measures

Correlations between the subscales of the CCC and the two alexithymia measures were examined. For the TD group, correlations were only run with CCC subscales that were reliable (see Methods). Given the number of correlations being run, the Benjamini and Yekutieli [31] correction was also applied; this correction is suitable for dependent tests such as in this case. Note that all correlations were two-tailed. Correlations are reported in Table 3. For the TD group, lower SR-alexithymia scores were associated with more pragmatic problems and general communication problems, although higher PR-alexithymia scores were associated with more pragmatic problems; however, after correcting for the number of correlations, these associations are no longer significant. For the DLD group, higher PR-alexithymia scores were associated with more pragmatic and general communication problems. Additionally, more speech problems were related to higher SR-alexithymia scores. The other structural language scales were not related to the PR-alexithymia scores. After applying corrections for multiple comparisons, the only correlation that remained significant was that between pragmatic language skills and PR-alexithymia in the DLD group.

For the DLD group, we also examined the associations between CCC scale scores and PR-alexithymia via a hierarchical regression, controlling for demographic variables (SES, gender, age and performance IQ). For the pragmatic subscale, we also examined the interaction between diagnostic group and pragmatic problems (because the other CCC subscales were not reliable in the TD group, this interaction effect could only be tested with this subscale). To protect against multicollinearity issues due to entering an interaction term, pragmatic subscale scores and diagnostic group membership variables were centred, and these centred variables were used to produce the interaction variable. Structural language subscales of the CCC never predicted PR-alexithymia. For pragmatics subscale, in the final step of this regression, the only significant predictors of PR-alexithymia scores were pragmatic problems, and the interaction between pragmatic problems and group (Table 4). This interaction indicates that the relation between more pragmatic problems and more alexithymia as reported with the PR-alexithymia was stronger in children with DLD compared to children without DLD.

### 3.4. Factor Structure of Parent-Reported Alexithymia Measures

We used an exploratory factor analysis on PR-alexithymia for both groups combined and separately, to examine whether multiple underlying constructs were being measured. If the PR-alexithymia measure (the CAM) not only measures alexithymia but also the general communication ability of children, more than one underlying factor would be present. We applied principal component analysis. Factor loadings and communalities are listed in Table 5.

When both groups were combined, the analysis indicated one strong factor (eigenvalue 7.56), which explained 54.0 percent of the variance. While for the TD data there were 2 Eigen values over 1, and 4 Eigen values over 1 for the DLD data, the three other metrics used to indicate the number of factors present in a dataset (parallel analysis, optimal coordinates, and acceleration factors) all indicated that the presence of just one factor in either sample. This one factor accounted for 44% of variance, in both DLD and TD samples.

## 4. Discussion

This study sought to examine the relationship between child and parent reports of alexithymic difficulties, and to consider whether the presence of language problems was associated with increased alexithymic difficulties. Previous literature has argued for a “language route” to the alexithymic profile, highlighting the need for further research into alexithymia in children with DLD [17,18]. However, disagreement between parent and child report measures of alexithymia has also been documented, but until now only in studies of children with autism or at genetic risk of autism [11], or in small samples which may have been underpowered to detect a correlation in the typical control sample [10].

We found that children with DLD scored higher on a measure of parent-reported alexithymia but not child-report measures. Similar to previous reports [10,11], parent and child report measures of alexithymia did not correlate with one another, in either children with or without DLD, or when the two groups were pooled together. Pragmatic language abilities were particularly related to parent-reported alexithymia scores, over and above membership of the DLD or TD group, though pragmatic language skills were more strongly related in the DLD group. Structural language problems however were not related to alexithymia scores in children with DLD.

Our report is the third and largest study, to our knowledge, to report non-significant or weak correlations between parent and child reports of alexithymia. We are also the first study to examine this agreement in DLD, rather than children with ASD or typical controls. These findings caution relying on only one source of information regarding children’s alexithymic traits, and suggests that these different reporters’ scores are reflecting different constructs. Indeed, the content of the measures is superficially quite different: the child report measure considers whether children can differentiate basic emotions and understand what caused them to feel an emotion, while our parent report measure reflects whether children communicate their emotions, or show incongruent emotional expressions and communication. Indeed, had we only collected parent or child measures of alexithymia, the comparison between DLD and TD children would have reached different conclusions. Using self-report measures, it appears children with DLD are not significantly more alexithymic than TD children; however, using parent-report measures, we would conclude that children with DLD are more alexithymic than their TD peers.

Given that we found non-significant associations between parent and child reports for both the TD and DLD groups, our results do not support the notion that a lack of agreement between child and parent report measures of alexithymia is an issue specific to children with developmental disorders, whom we might expect to have greater difficulty with self-insight. Rather, reports of alexithymia do not correlate even in typically developing children without communication problems.

Our findings offer partial support to the ideas expressed in Hobson et al. (2019), that the emotional difficulties reported in DLD may be explained through increased alexithymic difficulties in this group. The children with DLD themselves did not rate themselves as having higher alexithymic traits than their peers without DLD, but they increased scores relative to controls on parent-report measures of alexithymia, a measure which predominantly reflect problems with verbal expressions of emotions. However, we might have expected associations with structural language problems, if these language problems were contributing to children with DLD appearing to show alexithymic behaviours. This suggests that children with DLD do have alexithymic difficulties in expressing their emotions, beyond simply reflecting a language problem, and that apparent alexithymia in DLD is not simply measurement confound.

Another consideration is that parents completing a questionnaire on their children’s emotional abilities who report problems for the majority of the items (perhaps due to a general communication problem) may become biased to report increased difficulties on all items in the measure. However, the pattern of results cannot be fully explained by simple responder bias: if parents who felt their child had any sort of difficulty reported higher rates of problems on both communication and emotional measures, then it seems strange that pragmatic abilities specifically correlated with alexithymia, but not structural language abilities. Rather, pragmatic language abilities appear to be especially important for predicting parent-reported alexithymia. Intriguingly studies of communication problems and alexithymia following stroke in adults have also suggested that pragmatic rather than structural language problems may be particularly related to acquired alexithymic difficulties [23]. These findings provide some insight into what aspects of language and communication alexithymia may be associated to.

There are several avenues for future research to consider. Firstly, devising measures of alexithymia that are not verbally reliant would be most helpful for examining to what extent alexithymic difficulties in language impaired samples extend beyond problems with emotional expression. One possibility could be utilising physiological measures of emotional arousal as used in Gaigg, Cornell, and Bird [32]. These authors recorded galvanic skin responses while showing emotional images to their adult participants, and asked them to rate the strength of their emotional response to each trial. The correlation between self-ratings of emotional response and galvanic skin response itself correlated with self-report measures of alexithymia. This may suggest that physiological measures may provide some insight into the emotional responses of alexithymic individuals, but such an approach would require adaptation for children.

Secondly, the directionality of the association between parent-reported alexithymia and pragmatic difficulties has yet to be determined. Potentially the early social difficulties children with language problems face may restrict the amount of learning experiences children can have regarding their own and others’ emotions; thus, pragmatic language problems may lead to social problems, which may restrict emotional development. Alternatively, poor emotional insight may affect pragmatic abilities, as higher alexithymia may mean that inappropriate behavioural responses are selected in social situations, leading to increased ratings of pragmatic problems. Finally, pragmatic language skills and alexithymia may be jointly related to a third factor such the ability to recognise emotions in others or empathy.

Finally, there have been few investigations on the agreement between self-reports and other reports of alexithymia in adults, and thus it is unclear whether this issue pertains uniquely to developmental samples. One study with eating disordered women, aged between 13 and 31, did report positive, moderate correlations between self-reports and other reports of alexithymia [33]. This would seem to indicate that with older, predominantly adult samples, different sources of alexithymia reports correlate better. Alternatively, the sample of adult patients may have shown more elevated levels of alexithymia than our present developmental sample. This may mean, to the patients themselves or their parents, that alexithymic problems surpass a threshold of being noticeable and are thus reported. Perhaps when individuals’ emotional problems are subtle, there is less agreement between reporters. We would recommend a more systematic research programme investigating the agreement of different reporter measures of alexithymia, such that could better guide developmental researchers as to when to collect parent-report and when to collect child-report alexithymia measures. Research studies of adult dyads (e.g., romantic couples), where we can assume that all participants are past the developmental age at which they can reliably report on their alexithymia, may help illuminate to what extent disagreement in child–parent studies is due to developmental factors, or the nature of the alexithymia construct itself.

In addition to these outstanding questions, future researchers may also wish to address some of the limitations of our current design. Firstly, while we consider the issue of responder bias above, a better solution would likely be to have a third source of report, such as teacher or therapist report, so that correlations between parent-reported child communication difficulties and parent-reported child alexithymia could be considered in light of how another reporter views the child’s communication and emotional abilities. Behavioural measures of structural language abilities would also be helpful in untangling whether there is any support for the role of specific language processes in emotional insight: for example, we did not find associations between alexithymia and the semantic subscale of the communication checklist, but a standardised behavioural measure of vocabulary might provide a more specific, more valid index of a child’s vocabulary skills. Indeed, this paper utilized existing data that formed part on an ongoing programme of research, meaning that potentially informative variables were not all collected: for instance, future targeted research might make use of other sources of report, such as teacher report. Finally, this sample were recruited prior to a consensus building exercise and update to the criteria used to diagnose DLD [34], although we do not expect that this would have had a great impact on who was included in the DLD sample.

With regard to what the current study might recommend, not just for future research avenues but for professionals working with children with language disorder, our study highlights the need to be aware of the emotional abilities, including emotional expression abilities, of children identified as having language needs. Previous research has suggested that children with DLD show impairments in recognising emotions in others [35]; our results would suggest impairments in processing one’s own emotions as well. Such problems would have implications for the conduct of psychological therapy or interventions seeking to improve children’s wellbeing, if such interventions have adequate emotional expression skills as a prerequisite. Indeed, we know that children with language problems are over-represented and under-recognised in services for children with emotional and behavioural needs [36]: increased alexithymia may compound communication issues in these settings, reducing the accessibility and success of such interventions.

## 5. Conclusions

In summary, our study demonstrates a lack of association between parent and child measures of alexithymia, for both children with and without DLD. The reasons for this disagreement require further investigation, but the lack of agreement in the TD sample suggest previous reports of disagreement were not simply due to social-communication problems in the clinical samples. Child versus parent report measures may be capturing different aspects of the alexithymia construct, and parental measures of alexithymia may be particularly affected by language abilities. Indeed, children with DLD scored higher on parental measures of alexithymia. Pragmatic but not structural language abilities were related to parental reports of alexithymia, a finding that does not readily fit constructionist accounts of the role of language in emotion development but may highlight the importance of social skills and experience in learning about one’s own emotions.

## Figures and Tables

**Table 1 ijerph-18-08309-t001:** Participant Characteristics.

	Participant Group	
	TD (*n* = 183)	DLD (*n* = 106)
Age (years)	12.28 (1.41)	12.20 (1.92)
Age range	9.75–15.42	9.17–16.33
Gender	76 M, 107 F	55 M, 51 F
Neighbourhood SES	0.72 (0.95)	0.07 (1.07)
Performance IQ	107.18 (17.23)	93.89 (12.46)

SES = Socioeconomic Status.

**Table 2 ijerph-18-08309-t002:** Means and SDs for CAM items.

CAM Item	Participant Group	
	TD (*n* = 151)	DLD (*n* = 89)
When asked how he/she feels, answers with what they have done, instead of talking about feelings	1.73 (0.75)	2.24 (0.87)
Finds it difficult to say they feel unhappy while looking unhappy	1.54 (0.64)	2.06 (0.86)
Talks about unrelated topics instead of expressing their feelings	1.35 (0.57)	1.95 (0.82)
Has long periods with little emotional expression, interspersed with emotional outbursts	1.13 (0.41)	1.44 (0.68)
Finds it difficult to say that they feel happy while looking happy	1.23 (0.52)	1.65 (0.72)
Walks away when asked to talk about feelings	1.37 (0.58)	1.91 (0.95)
Is incoherent when asked to talk about feelings	1.34 (0.60)	2.19 (0.85)
What they say about feelings does not match the feelings they show	1.22 (0.47)	1.61 (0.72)
Changes topic of conversation when asked to talk about feelings	1.43 (0.57)	1.99 (0.84)
Has difficulty naming positive emotions (such as joy, happiness or excitement)	1.26 (0.51)	1.75 (0.84)
Says “forget it” or “leave me alone” when asked how they feel	1.50 (0.65)	1.98 (0.90)
Has trouble finding the right words or can’t get out their words when they talk about own feelings	1.45 (0.66)	2.44 (0.82)
Uses few words (e.g., only “good”/“bad”) to describe most of their feelings	1.71 (0.86)	2.55 (0.94)
Says “I don’t know” when asked why they are upset	1.64 (0.73)	2.37 (1.00)

All comparisons yielded *p* < 0.001 (two-tailed). TD = Typically Developing; DLD = Developmental Language Disorder.

**Table 3 ijerph-18-08309-t003:** Correlations between parent and child-report alexithymia measures and subscales of the CCC.

	TD		DLD	
CCC Scale	PR-Alexithymia (CAM)	SR-Alexithymia (DIF)	PR-Alexithymia (CAM)	SR-Alexithymia (DIF)
Speech	-	-	0.08	0.23 *
Syntax	-	-	0.12	0.09
Semantics	-	-	0.14	0.19
Coherence	-	-	0.11	0.20
Pragmatics	0.20 *	−0.20 *	0.41 **^,†^	0.10
General	0.14	−0.18 *	0.31 **	0.19

All significance values are two-tailed. * *p* < 0.05, ** *p* < 0.001, ^†^ survived Benjamini and Yekutieli (2001) correction. Note that as several subscales of the CCC were not internally reliable in the TD sample, correlations with these subscales are not reported. CCC = Children’s Communication Checklist.

**Table 4 ijerph-18-08309-t004:** Hierarchical regression results for CAM scores.

	Model 1		Model 2		Model 3	
	*R*^2^ = 0.27, *F* for *R*^2^ Change *=* 80.77		*R*^2^ = 0.33, *F* for *R*^2^ Change = 19.09		*R*^2^ = 0.35, *F* for *R*^2^ Change = 6.22	
Variable	Beta	*p*	Beta	*p*	Beta	*p*
SES	−0.13	0.04	−0.11	0.07	−0.09	0.12
Gender	−0.06	0.33	−0.07	0.25	−0.07	0.20
Age	0.01	0.89	0.02	0.77	0.02	0.74
Performance IQ	−0.04	0.55	−0.03	0.67	−0.03	0.60
Group	0.45	<0.001	0.19	0.04	0.11	0.28
Pragmatics			0.35	<0.001	0.37	<0.001
Pragmatics × Group					0.14	0.03

Diagnostic group, pragmatics and pragmatics × group variables are centred to reduce multicollinearity.

**Table 5 ijerph-18-08309-t005:** Communalities and factor loadings for CAM items.

	DLD			TD			Whole Sample		
		Component Factor Loadings			Component Factor Loadings			Component Factor Loadings	
	Communality	1	2	Communality	1	2	Communality	1	2
When asked about how feeling, instead talks about what has been doing	0.318	0.527		0.249	0.471		0.331	0.559	
Has difficulty saying feels sad, even through looks sad	0.510	0.701		0.467	0.683		0.577	0.726	
Talks about unimportant things/topics instead of sharing feelings	0.551	0.702		0.639	0.759		0.662	0.779	
Has long periods of little/no emotional expression, interrupted by bursts of emotional expression	0.627	0.422	0.670	0.457	0.633		0.397	0.569	
Has difficulty saying they’re happy even though looks happy	0.553	0.741		0.632	0.666	0.435	0.678	0.736	
Physically removes self from situations when asked to talk about feelings	0.479	0.684		0.775	0.708	−0.524	0.788	0.732	−0.503
Makes up unrelated stories when asked about their feelings	0.480	0.689		0.574	0.756		0.667	0.788	
Verbal expressions of feelings do not match non-verbal expressions	0.652	0.748		0.560	0.682		0.577	0.743	
Changes the topic of conversation when asked about their feelings	0.736	0.784		0.586	0.747		0.650	0.799	
Has difficulty naming their positive feelings (such as joy, happiness, excitement)	0.768	0.835		0.347	0.587		0.592	0.765	
Says “forget it” or “leave me alone” when asked about their feelings	0.545	0.616	0.408	0.697	0.656	−0.517	0.573	0.676	
Has trouble finding words or getting words out when talking about their own feelings	0.490	0.698		0.616	0.777		0.657	0.805	
Uses few words (may just say “good”/“bad”) to describe most of their feeling	0.430	0.645		0.556	0.733		0.610	0.751	
Says “I don’t know” when asked why he/she is upset	0.688	0.781		0.581	0.750		0.665	0.804	

## Data Availability

Data is available upon request by contacting N.P.v.d.B.

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
