# Peer review of "The Association between Parent and Child-Report Measures of Alexithymia in Children with and without Developmental Language Disorder"

_ijerph, 2021, doi:10.3390/ijerph18168309_

Round 1

Reviewer 1 Report

Dear Authors, thank you for submitting the revised version of your manuscript,  for your reply to my comments and the related additional information in the manuscript, especially regarding the methods of the study. The manuscript has improved a lot and is almost good to go. 

Authors now specified that the study has been conducted in 2014. Could you please explain it hasn't been published before.  I am also wondering if this constitutes a limitation that should be acknowledged in the limitation section. 

Reviewer 2 Report

The authors should include an explanatory figure of the model proposed in the introduction and the values that the study shows of the variables in the results section, thus building an SEM.
On the other hand, I don't understand what is the reason for including item analysis? this should be eliminated and the analyzes focus on the verification of the SEM proposed in the introduction.

Author Response

This manuscript is a resubmission of an earlier submission. The following is a list of the peer review reports and author responses from that submission.

Round 1

Reviewer 1 Report

The authors present an interesting study, with an important topic and they describe quite interesting results. It was my pleasure to review your paper, it was really interesting and valuable. I would like to highlight a few points as recommendations and comments.

Introduction

The introduction is well written and presented. However, I consider that the knowledge gap that fills this work should be highlighted

Methods

It is necessary to include the analysis plan as a section, and in it, a detailed description of the multivariate analyses.

Results

Due to the differences between samples depending on the SES and Performance IQ, these variables must be controlled in multivariate analyses. It would be advisable to separate the explained variance from these variables.

Conclusions

I do not see what the manuscript contributes, as other studies come to the same conclusions.
The authors should make more concrete recommendations related to the conclusions.

Misprints

I have not detected any misprints

Author Response

We thank the reviewer for their time reviewing our paper. Below we list their points, with our responses/summary of changes in red text. 

***

The introduction is well written and presented. However, I consider that the knowledge gap that fills this work should be highlighted

We thank the reviewer for this comment, and agree that the introduction would benefit from explicitly stating what knowledge gap our paper seeks to address. We have added lines to this effect, at the end of the introduction (Paragraph starting “Overall, there has been limited study…”).

Methods - It is necessary to include the analysis plan as a section, and in it, a detailed description of the multivariate analyses.

We apologise for this oversight – we have now added the “analytical approach” section at the end of the Method. Much of the required content was presented at the start of the Results section, and hence we have re-used some elements that were previously presented at the start of the Results, but expanded upon them more fully, including how we conducted our MANCOVA.

Results - Due to the differences between samples depending on the SES and Performance IQ, these variables must be controlled in multivariate analyses. It would be advisable to separate the explained variance from these variables.

We thank the reviewer for this comment. In revisiting the analyses, we decided that a better approach for comparing the DLD and TD groups would be to conduct a MANCOVA, with SES and PIQ entered as covariates to control for these group differences when looking at the alexithymia scores of the samples. In order to quantify the variance explained by group membership versus these background variables, we have also now conducted a hierarchical regression and report the r square change. This shows that group (i.e. DLD or TD group membership) predicts an additional 15% of variance in PR-alexithymia, over and above PIQ and SES. These analyses replace the t-tests reported in our original manuscript.

Conclusions - I do not see what the manuscript contributes, as other studies come to the same conclusions.

Indeed, two previous studies had reported a lack of a correlation/weak correlations between parent and child alexithymia reports. However, these studies were concerned with autism. Hobson et al (2020)’s sample were a sample of adolescents who had autism or whose twin had autism (and thus were themselves part of the broader autism phenotype), while the Griffin paper had a sample of children with ASD and a TD control sample, but these separate samples were quite small, too small to detect a correlation between the reports unless it was quite large. Thus, our paper examines alexithymia in a new group, extending the previous work on ASD. The examination of alexithymia in DLD is also novel, and called for, given the "language hypothesis" of alexithymia. The additional elements at the start of our Discussion now make clearer what our study adds.

The authors should make more concrete recommendations related to the conclusions.

 We thank the reviewer for this comment. We make a few recommendations for future research, so we take this comment to mean recommendations for practice. With regards to recommendations for practice, at present we feel we should be a bit hesitant about recommending the widespread use of alexithymia measures when working with children with DLD (for the reason that the lack of agreement between our child self-report and parent report measures means it is as yet unclear whether one is a more valid measure than the other, or whether they are reflecting each only part of the alexithymic construct). However, the significant group differences on the parent report measure would indicate that children with DLD appear to have problems expressing their own emotions. Combined with the fact that previous research has shown an overrepresentation of children with language needs in mental health settings, one implication of our study for clinical practice would be the potential impact on psychological therapy. We outline this in our Discussion section, prior to our conclusions.

Reviewer 2 Report

Thanks for letting me review this work. The work seeks to investigate the association between child self-report and parent report measures of alexithymia, and examine their associations with communication difficulties. Specifically, we tested: a) whether children with DLD would score higher on either self-report and parent-report measures of alexithymia compared to children without DLD; b) whether parent and child report measures would correlate, for children with and without DLD; c) whether pragmatic versus structural language problems would be associated with alexithymia in children with and without DLD; d) whether measures of alexithymia that show relationships to language abilities show a similar factor structure for children with and without DLD.

Seen like this, it seems an interesting work, but it presents great problems at a methodological and statistical level:

The sample of parents who complete the questionnaire is not described. The responses of the children and those of the parents are included in the same subject. They are different subjects so they cannot be viewed as the same sample.
It lacks the data analysis section.
The formulation of the objectives is not correct, the objectives are formulated in the infinitive. In addition, they are not answered with the data analyzes performed.
The rationale is very weak, without indicating the theory behind the analysis or the definitions on which the analysis of the subsequent constructs is based.
This also limits the discussion to repeating the study results.

Author Response

We thank the reviewer for their time reviewing our paper. Below we list their points, with our responses/summary of changes in red text. 

***

The sample of parents who complete the questionnaire is not described. The responses of the children and those of the parents are included in the same subject. They are different subjects so they cannot be viewed as the same sample.

The children are reporting on their own alexithymia and the parents are reporting on their children’s alexithymia (not their own alexithymia). The subject is still the child, but the sources of alexithymic report differ. This has been the approach in other alexithymia studies in which parent and child reports were compared (e.g. Griffin et al, Hobson et al., as cited in the manuscript) and in other studies that examined agreement for factors such as anxiety and depression. Nonetheless, we have added some additional information in the participant section with regards to the parents’ own characteristics.

It lacks the data analysis section.

We apologise for this oversight – we have now added a section “analytical approach” at the end of the Method. Much of the required content was presented at the start of the Results section, and hence we have re-used some elements that were previously presented at the start of the Results, but expanded upon them more fully.

The formulation of the objectives is not correct, the objectives are formulated in the infinitive. In addition, they are not answered with the data analyzes performed.

We have rephrased the elements at the end of our introduction to address the tensing errors, and to align our objectives and our analyses. We have also reordered the results section and activities a-d (listed at the end of our introduction) so that the order in which we do things is consistent throughout the manuscript, hopefully helping to more clearly illustrate how our objectives and research questions align with the analyses that we performed.

The rationale is very weak, without indicating the theory behind the analysis or the definitions on which the analysis of the subsequent constructs is based. This also limits the discussion to repeating the study results.

We thank the reviewer for highlighting the need for tighter links to psychological theory and to make more explicit the study rationale. Given the number of analyses in our Results section, our feeling is that reminding readers of the key findings at the start of the Discussion is helpful before unpacking the implications of these findings; however, we now open our Discussion section with a quick revisitation to some key material in the introduction. Namely, that there has been proposed a “language route” to alexithymia, which would propose that children with DLD should be more alexithymic, and that there have been previous reports of no/weak correlations between parent and child alexithymia reports, but that these studies have been on children with autism (and the TD groups too small to show whether in TD children there is a correlation between parent and child reports).

Reviewer 3 Report

Thank you for providing me the opportunity to review this manuscript, which reports on the association between parent and child-report of Alexithymia.

The topic of the study has clinical and public health importance, and is certainly in need of high-quality research. Overall, the manuscript is well-written. The introduction makes a case for the current study, the methods seem appropriate and the results are reported in a consistent way. Below are a few minor comments that may help modifying a few unclarities in the paper.

Materials and Methods

Please specify when recruitment occurred (authors referred to DSM-IV criteria).

Specify the statistical packages used for this study.

Please include sample/power calculation.

Table 2 is difficult to read. Could you align the items to the left margin?

Section 3.4. Factor structure: What type of factor analysis has been conducted. Please motivate and provide psychometric data.

Section 4, Discussion: Please include a paragraph regarding the limitations of the study.

I hope that these few comments may help the authors revising the manuscript. Good luck.

Author Response

We thank the reviewer for their time reviewing our paper. Below we list their points, and give our responses/summary of changes in red text.

***

Materials and Methods - Please specify when recruitment occurred (authors referred to DSM-IV criteria).

Thank you for this comment. This is now detailed in the Participants subsection.

Specify the statistical packages used for this study.

We now detail this at the end of our analytical approach section, at the end of the Methods.

Please include sample/power calculation.

We have added this consideration to our analytical approach section (copied below, for ease). For transparency, the nature of the project was a large ongoing research programme with a developmental sample and power calculations for these specific tests were not conducted prior to data collection. However, our dataset is sufficiently large to test our key hypotheses (detailed below).

“Readers may note that the data reported here formed part of a wider project on DLD [18, 20], and thus power calculations for the specific analytical tests detailed here were not conducted prior to data collection: however, we did consider whether our dataset would be sufficient for our main objectives. We posit that to argue that parent and child reports of alexithymia were in agreement, we would expect at least a moderate sized correlation; this would require 88 parent-child datapoints, based on power calculations in G*Power [27], based on 90% power. Our sample size was thus sufficiently large to allow us to explore this separately in the two groups (TD and DLD). Similarly, a MANCOVA comparing the two groups on alexithymic traits, assuming a medium effect and 90% power, would require a total sample size of 171 participants. Thus, we were suitably powered to detect meaningful relationships between parent and child alexithymic reports, and differences between DLD and TD groups.”

Table 2 is difficult to read. Could you align the items to the left margin?

 Yes, we agree that the formatting here was problematic. This has been amended.

Section 3.4. Factor structure: What type of factor analysis has been conducted. Please motivate and provide psychometric data.

We have now added these details to this section, and factor loadings and communalities are reported in the new Table 5 (with the different results for the TD-only, DLD-only and whole sample analyses demarcated).

An EFA was conducted, as we were interested in whether the alexithymia measures would show a factor structure that suggested components reflective of language ability, but we did not have set ideas about how many factors might emerge. We detail this rationale in our analytical approach section.

Section 4, Discussion: Please include a paragraph regarding the limitations of the study.

We apologise for this oversight – we now include a limitations paragraph, after our recommendations for future research, including noting that our study would have benefited from a third source of report (to rule out possible responder bias) and behavioural measures of structural language.